# Genome Survey of Male and Female Spotted Scat (*Scatophagus argus*)

**DOI:** 10.3390/ani9121117

**Published:** 2019-12-11

**Authors:** Yuanqing Huang, Dongneng Jiang, Ming Li, Umar Farouk Mustapha, Changxu Tian, Huapu Chen, Yang Huang, Siping Deng, Tianli Wu, Chunhua Zhu, Guangli Li

**Affiliations:** Guangdong Research Center on Reproductive Control and Breeding Technology of Indigenous Valuable Fish Species, Fisheries College, Guangdong Ocean University, Zhanjiang 524088, China; 15766242759@163.com (Y.H.); jdn1987@163.com (D.J.); lm1039931841@163.com (M.L.); umarfk.gh@gmail.com (U.F.M.); tiancx@gdou.edu.cn (C.T.); chpsysu@hotmail.com (H.C.); zjouhy@126.com (Y.H.); sipingdeng@126.com (S.D.); wtianli@163.com (T.W.); zhu860025@163.com (C.Z.)

**Keywords:** genome size, illumina sequencing, *Dmrt1*, sex determining gene, aquaculture, sex control

## Abstract

**Simple Summary:**

The spotted scat, *Scatophagus argus*, is a marine aquaculture fish species that is economically important in Asia. As the spotted scat exhibits notable sexual dimorphism with respect to growth, aquaculture efficiency can be increased through the practice of sex control breeding. However, genomic data from *S. argus* is lacking. In the present study, a genomic survey was conducted using next-generation sequencing technologies. Data, including the size of the genome, sequence repeat ratio, heterozygosity ratio, whole genome sequence and gene annotation were obtained. This information will serve to support the breeding and aquaculture of *S. argus*.

**Abstract:**

The spotted scat, *Scatophagus argus*, is a species of fish that is widely propagated within the Chinese aquaculture industry and therefore has significant economic value. Despite this, studies of its genome are severely lacking. In the present study, a genomic survey of *S. argus* was conducted using next-generation sequencing (NGS). In total, 55.699 GB (female) and 51.047 GB (male) of high-quality sequence data were obtained. Genome sizes were estimated to be 598.73 (female) and 597.60 (male) Mbp. The sequence repeat ratios were calculated to be 27.06% (female) and 26.99% (male). Heterozygosity ratios were 0.37% for females and 0.38% for males. Reads were assembled into 444,961 (female) and 453,459 (male) contigs with N50 lengths of 5,747 and 5,745 bp for females and males, respectively. The average guanine-cytosine (GC) content of the female genome was 41.78%, and 41.82% for the male. A total of 42,869 (female) and 43,283 (male) genes were annotated to the non-redundant (NR) and SwissProt databases. The female and male genomes contained 66.6% and 67.8% BUSCO core genes, respectively. Dinucleotide repeats were the dominant form of simple sequence repeats (SSR) observed in females (68.69%) and males (68.56%). Additionally, gene fragments of *Dmrt1* were only observed in the male genome. This is the first report of a genome-wide characterization of *S. argus*.

## 1. Introduction

The spotted scat, *Scatophagus argus*, (Perciformes, Scatophagidae), is a popular species of fish known for both its aesthetic value and human consumption due to its rhombic spotted body and high nutrient value [1,2,3]. Moreover, unlike other species, the cultivation of *S. argus* is relatively easy with low cost of feeding and high market price, making it an important farmed fish with considerable economic value in East and Southeast Asia [4,5,6,7]. The *S. argus* is widely distributed in the littoral and salt/freshwater rivers of the Indo Pacific, Australia, the Malay Archipelago, the Philippines and South and South East Asia, including China [8]. Analysis of the gut contents of *S. argus* has revealed a combination of both algae and detritus, indicating that it is an omnivorous fish [8]. *S. argus* is able to tolerate movement directly from freshwater to seawater, suggesting that it has a robust capacity for osmoregulation [9]. Furthermore, *S. argus* is a leading seafood for both its desirable taste and high nutritional value [1,8]. The adaptability of *S. argus* to a broad range of temperatures and levels of salinity, combined with their excellent edibility, has enabled it to become an attractive aquaculture species in China. However, the main supply of *S. argus* fingerlings derives from wild capture. In the absence of proper management, this has the potential to endanger the resource through overfishing [10]. Most studies examining this species have focused on reproduction and the importance of solving challenges associated with artificial propagation [11,12,13,14]. The female *S. argus* grows significantly faster and larger than males. Therefore, from an economic perspective, all-female farms will improve the rate of production and the total market value. Thus, sex control would be a good strategy to employ in an *S. argus* breeding program. In addition, understanding the mechanisms of sex determination and differentiation would be crucial to the effective maintenance of all-female production [15]. In addition, differences in growth rate between males and females provide a valuable model to explore the mechanisms of sexual dimorphisms in vertebrates [16]. To date, the genomic information of *S. argus* is lacking, essential for basic and applied research of this species [17].

Next-generation, high-throughput sequencing (NGS) is an efficient strategy for generating genomic resources. This technology is currently in wide use for transcriptomic and genomic studies [18,19,20,21]. Many genes in *S. argus* related to reproduction have previously been identified from the transcriptome of mixed ribonucleic acid (RNA) from various female and male tissues [4]. Furthermore, a comparative transcriptomic analysis of testicular and ovarian tissue has discovered many genes involved in sex determination and differentiation in *S. argus* [22]. Recently, NGS was used to conduct genomic surveys which enhanced the field’s understanding of genetic variation, evolutionary analysis, genome structure analysis, and marker development [23,24,25]. To further compare the genomes of male and female *S. argus*, complete genome sequences were obtained using NGS, the data of which was used to assemble the genome, perform genome size estimation, evaluate the guanine-cytosine (GC) content and identify simple sequence repeats (SSR). These data will be the basis of a fundamental genomic resource for reproduction-related studies. In addition, these data also provide a foundation for future genomic studies of *S. argus*.

## 2. Materials and Methods

### 2.1. Sample and Tissue Collection

Specimens of *S. argus* were obtained from the Zhanjiang Donghai Island Cultivation Base (Zhanjiang, Guangdong, China). Two adults, one female and one male, were subjected to genome sequencing. The animals were immediately dissected following tricaine MS-222 anaesthesia. White muscle tissue was used for DNA extraction. Samples were flash-frozen in liquid nitrogen for 1 h before storage at −80 °C. All animal experiments were conducted in accordance with the guidelines and approval (201903004) of the Animal Research and Ethics Committees of the Institute of Aquatic Economic Animals of Guangdong Ocean University.

### 2.2. Whole Genome Sequencing

Genomic DNA was isolated from muscle using a nucleic acid purification kit (N1173, DONGSHENG BIOTECON, Guangzhou, China) according to the manufacturer’s instructions. The DNA was then sheared randomly into small fragments using an ultrasonic shearing device. Two paired-end libraries with an insert size of 350 base pairs (bp) were constructed from randomly fragmented genomic DNA, following a standard protocol (Illumina, Beijing, China). The DNA library was then sequenced in paired-end, 150-bp mode using the Illumina HiSeq X Ten platform (Novogene, Beijing, China) in accordance with the manufacturer’s instructions.

To obtain clean reads, raw reads were filtered using the high-throughput quality control (HTQC) package [26]. The raw data were cleaned as follows: (1) adaptor sequences introduced during sequencing library construction were removed; (2) paired reads were removed when at least 10% of nucleotides were uncertain in either read; (3) paired reads were discarded when low-quality nucleotides (base quality <5) accounted for >50% of either read. Next, 5,000 clean-read pairs from each library were randomly selected and blasted against the National Center for Biotechnology Information (NCBI) nonredundant (NR) nucleotide database to check for obvious sample contamination. All subsequent analyses were based on these clean reads. Entire read sets were deposited in the short read archive (SRA) databank (http://www.ncbi.nlm.nih.gov/sra/) and are available under accession number PRJNA559409.

### 2.3. Genome Size Estimation and Identification of Heterozygosity and Repeat Ratios

A K-mer analysis was performed for each library to estimate the genome size, level of heterozygosity and repeat frequencies of the genomes by Marçais [27] to assess the genome complexity of *S. argus*. The K-mer statistic was used to assign discrete probability distributions for a number of possible K-mer combinations [28]. To minimize the influence of sequencing errors, low-frequency K-mers (≤5) were discarded. The copy number of a given K-mer (17-mer) present in all clean Illumina reads were counted then divided by the total length of each sequence read. The distribution of copy numbers was then plotted. The K-mer distribution can be used to infer the size of the genome. The peak value of the frequency curve represents the overall sequencing depth. Genome size was calculated as follows: K-mer number/peak depth [27], Revised genome size = genome size × (1 − error rate).(1)

In a heterozygous genome, the single nucleotide polymorphism (SNP) sites will be sparse, and ideally 2 × K heterozygous K-mers around each SNP site would be present. Heterozygous K-mers will have half of the expected coverage depth compared to the homozygous K-mers. The heterozygosity rate can be estimated as follows: Heterozygosity rate = (a_1/2_ × n_Kspecies_/(2 × K))/(n_Kspecies_ − a_1/2_ × n_Kspecies_/2) = a_1/2_ /(K(2 − a_1/2_)).(2)

In this formula, n_Kspecies_ denotes the total number of K-mer species and a_1/2_ denotes the ratio of heterozygous K-mer species [29,30]. The difference between the k-mer distribution and Poisson distribution is large due to the presence of incorrect sequences, or the number of sequence layers, which affects later estimates. Hence, the repeat rates were calculated according to the percentage of the total number of K-mers after the main peak, which was 1.8 times of all K-mer numbers [29].

### 2.4. Genome Assembly and GC Content

The software SOAPdenovo (v2.04) was used for de novo genome assembly [31]. All clean reads were used in the assembly, with a K-mer size of 41 selected as the default parameter used to construct a de Bruijn graph [32]. The de Bruijn graph was simplified by breaking the connections at repeat boundaries, with unambiguous sequence fragments outputted as contigs. After realigning all useable reads to the contig sequences and obtaining aligned paired-end reads, the number of shared paired-end reads having a relationship between each pair of contigs was calculated, the rate of consistent and conflicting paired-end reads was rated and scaffolds constructed step-by-step using SOAPdenovo. Gaps in the scaffolds were closed using GapCloser software (v1.12), and those longer than 100 bases in length were selected.

To measure the sequencing bias of *S. argus*, the guanine plus cytosine (GC) content and average sequencing depth were counted. The GC content is strictly controlled and moderately balanced across the genome [33,34]. Ten kb non-overlapping sliding windows were used along the assembled sequences to calculate the GC average sequencing depth.

### 2.5. Gene Prediction, Annotation, and Assembly Assessment

GlimmerHMM software (v.3.01) was used for de novo prediction of genes using default parameters (genemodel = zebrafish) [35]. Next, the predicted genes were used to BLAST the NR and SwissProt databases using BLASTx (E-value < 1e−5). Gene ontology (GO), Clusters of euKaryotic Orthologous Groups (KOG) and Kyoto Encyclopedia of Genes and Genomes (KEGG) pathway annotations were also assigned to genes using Blast2GO software [36]. In addition, the genes described in this way were classified into KOG slim and GO categories, then mapped onto the KEGG descriptors. To assess the completeness of the assembly, a Benchmarking Universal Single-Copy Orthologs (BUSCO, v.3.0) evaluation was performed using the Actinopterygii_odb9 database [37].

### 2.6. Alignment of The Male and Female Scaffolds on Dmrt1s Exons

Our previous research indicated that *Dmrt1* is linked to the Y chromosome, and the truncated homologue *Dmrt1b*, is X chromosome-linked [38]. The predicted *Dmrt1* and *Dmrt1b* genes observed during *S. argus* transcriptome analysis were mapped to the draft genome using Localblast software (NCBI-blast-2.2.27) [22]. Sequence homology alignment of male and female fragments was performed using the MegAlign application of DNASTAR software (http://www.dnastar.com).

### 2.7. Identification of SSRs

Sequence repeat search software in the MIcroSAtellite (MISA) model was used to detect simple repeat sequences (SSRs) in the DNA sequence. The software is divided into two modules. The first module was used to detect all the SSRs in the DNA sequences. The minimum numbers of SSRs for mono-, di-, tri-, tetra-, penta- and hexa-nucleotides adopted for identification were 10, 6, 5, 5, 5, and 5, respectively. The second module was used to filter the results of the first module and then remove SSRs which were too close.

## 3. Results

### 3.1. Genome Sequencing and Sequence Quality Estimation

A total of 55.809 and 51.154 GB of raw data were generated from female and male *S. argus*, respectively. After filtering, 55.699 (female) and 51.047 (male) GB of clean data were obtained, with Q30 scores assigned to 91.94% and 92.03% of the female and male libraries, respectively. The error rates of these libraries were 0.03% (Table 1). It was observed that the best BLAST results of reads were enriched for closely related fish species, including *Dicentrarchus labrax*, *Scatophagus argus*, *Haplochromis burtoni* and *Oreochromis niloticus* (Appendix A).

### 3.2. Genome Size, Ratio of Heterozygosity, and Repeats by K-Mer Analysis

The K-mer analysis indicated that the depth of female and male *S. argus* were at 74 and 68×, respectively (Figure 1, Table 2). The estimated size of the female and male genomes was 613.16 and 612.32 MB, respectively. The revised genome sizes were 598.73 and 597.60 MB, respectively. The rates of heterozygosity were calculated to be 0.37% and 0.38%, and repeat rates calculated to be 27.06% and 26.99%.

### 3.3. Genome Assembly and GC Content

According to the analysis, a total of 444,961 and 453,459 contigs were assembled from female and male *S. argus*, with an N50 of 5747 bp and 5745 bp, respectively. Based on the contigs, the genome assembly contained 335,162 and 340,134 scaffolds, with an N50 of 13,556 and 13,591 bp (Table 3). The average GC content of female and male *S. argus* genomes were 41.78% and 41.82%, respectively. In Figure 2, the red regions represent relatively dense portions of the scatter plot, with the GC depth being blocked into two layers, partly due to the rate of heterozygosity [39].

### 3.4. Gene Prediction, Annotation, and Evaluation

Based on our assembled genome sequences, a total of 94,862 female genes and 95,273 male genes were predicted by GlimmerHMM software. The predicted genes ranged from 101 to 52,424 bp in length. Among the predicted genes, 42,869 female and 43,283 male genes were functionally annotated in the NR and SwissProt databases (Table 4). The KOG, KEGG and GO annotation or classification of these annotated genes were similar in the males and females (Appendix A). Among 4584 conserved Actinopterygii genes searched, 3055 (66.6%) and 881 (19.2%) BUSCO core genes were completed and partially identified, respectively, in the female genome. The BUSCO core genes of the male genome were similar to those of the female (Appendix A). The partially identified core genes in both sexes were a little high caused by incomplete genome assembly based on short NGS reads.

### 3.5. Characterization of the Dmrt1s Gene

Gene prediction and annotation confirmed that *Dmrt1* is male-specific, whereas *Dmrt1b* is observed in both male and female fish. Three and four scaffolds containing *Dmrt1b* and *Dmrt1* were obtained from the female and male genomes, respectively (Figure 3, Appendix A). Alignment demonstrated that *Dmrt1* exons 1, 2, 3, and 4 had 79.9%, 90.7%, 75.8%, and 84.1% similarity with the corresponding fragments on *Dmrt1b*, respectively. The *Dmrt1b* fragment corresponding to *Dmrt1* exon 5 was not found, possibly be due to incomplete sequencing in female fish. The female scalffold49157 had a similarity of 70.2% with the *Dmrt1* 3’ UTR on the male scalffold152219 in their overlapping region (~500 bp). In the first and second exons of *Dmrt1b*, several mutations were observed that resulted in the premature termination of *Dmrt1b*. In addition, five male scaffolds (162,645, 106,307, 107,747, 68,937, and 118,347) were found to cover the *Dmrt1b* region (Appendix A). The average length of scaffolds covering *Dmrt1b* from male and female are 1448 and 3667 bp, respectively. It seems that the assemble quality of *Dmrt1b* region in male is relatively lower than that of the female.

### 3.6. Identification of SSR

After filtering the SSR sequences from the contig sequences at both sides (the distances of SSR sequences to contig sequences at both ends were less than 100 bp), a total of 299,574 and 299,893 SSRs were detected in female and male fish, respectively (Table 5). In the female, the predominant SSR motif types observed were dinucleotide repeats, occurring at 205,789 loci (68.69%), followed by trinucleotide repeats (31,228, 10.42%) (Figure 4). Among the dinucleotide repeat motifs, the AC/GT repeats were the most abundant, accounting for 75.85% (Appendix A). The most common tri-nucleotide motifs were AGG/CCT and AAT/ATT, accounting for 31.36% and 27.53%, respectively (Appendix A). In the male fish, the SSR motif profile was similar to that of the female (Figure 4; Appendix A; Table 5).

## 4. Discussion

According to K-mer (K = 17) analysis, the sizes of female and male *S. argus* genomes were estimated to be 598.73 MB and 597.60 MB, respectively. It has been reported that using a bulk fluorometric assay, the DNA content of *S. argus* red blood cells is approximately 0.77 pg [40]. According to a mass conversion formula, 1 pg DNA equates to approximately 0.978 × 10^9^ bp. Based on this estimation, the genome size of *S. argus* is approximately 753.06 MB, which is larger than the K-mer predicted genome size [41]. The larger genome size estimated by the fluorometric assay could be due to the non-specific nature of the fluorescent dye used, possibly detecting binding to non-genomic nucleotides. The size of the genome of *S. argus* estimated in this study was larger than that of *Sillago sinica* (534 MB) [42], but smaller than that of *Lateolabrax maculatus* (670 MB) [43], *Dicentrarchus labrax* (675 MB) [44] and *Larimichthys crocea* (679 MB) [45].

The rate of heterozygosity in the female was 0.37%, which is lower than that of the male fish (0.38%). Consistently, sex-linked markers demonstrated that the sex-determination system of *S. argus* is male heterozygous XY and female homozygous XX. This indicates that the heterozygosity rate of XY males should be higher than that of female fish [38]. However, the heterozygosity rate difference between the sexes was only 0.01%. This is possibly because the Y chromosome is still “young” in *S. argus* [46]. Consistently, no morphologically distinguishable sex chromosome has been observed in the species [47]. Because lower rates of heterozygosity tend to simplify genome assembly, the data presented here suggest that female *S. argus* would be preferable to males for the development of a draft genome in future studies [28]. The heterozygosity rate observed here was higher than that of *Oplegnathus fasciatus* (0.29%) [48], but smaller than that of *Pelteobagrus fulvidraco* (0.45%) [49], *Seriola dumerili* (0.65%) [50] and *Sillago sinica* (0.66–0.76%) [42]. The lower heterozygosity rate of *S. argus* suggests that wild-caught fish are close to being over-fished.

The genome assembled here appears to be of higher quality than that of *S. sihama* in spite of the same sequencing strategy being employed, also in our lab [51]. This might be due to the heterozygosity rate of *S. argus* being substantially lower than that of *S. sihama* (0.92%) [51]. Male-specific *Dmrt1* is the candidate sex determination gene in *S. argus* [38]. Consistently, gene fragments of *Dmrt1* were present in the male genome (Appendix A), while being absent in the female genome (data not shown). The complete *Dmrt1* gene sequence is not available at present due to the read quality and ultimately, assembly status. On the other hand, the *Dmrt1* gene is always very long in fish species. For example, the shortest *Dmrt1* gene was found to be 12kb in *Takifugu rubripes*, which has a compressed genome [38]. To obtain better-assembled genomes by NGS, long-insert libraries should be constructed for sequencing [20]. Alternatively, third-generation sequencing technologies, such as Pacific Biosciences (PacBio) sequencing platform could be used to enhance genome assembly in future studies [42].

Both male and female genomes appeared capable of developing tremendous SSR markers, which will help to solve the problem that SSR markers are principally derived from transcriptomic data in *S. argus* [4,52]. The number of SSR markers observed in male fish was slightly higher than that of female fish. The difference might be due to the male proto-Y sex chromosome having more repetitive elements [38,53].

## 5. Conclusions

In the present study, the first reference genome of *S. argus* was sequenced. Two spotted scat, a female and male, underwent whole-genome sequencing. The genome sizes of the female and male were 598.73 MB and 597.60 MB, respectively. The genome was annotated with 42,869 female and 43,283 male genes. The *S. argus* genomes contained 66.6% and 67.8% of the core genes in conserved Actinopterygii orthologs for each sex, respectively. The rate of genome heterozygosity of the male fish was slightly higher than that of the female. The number of SSR markers developed from the male was slightly greater than that of the female fish. These data suggest that the differences between male and female genomes of *S. argus* are minor. It was also confirmed that the male-specific *Dmrt1* is a good candidate sex determination gene via genome sequencing. This study provides an important genome resource for further studies of *S. argus*.

## Figures and Tables

**Figure 1 animals-09-01117-f001:**
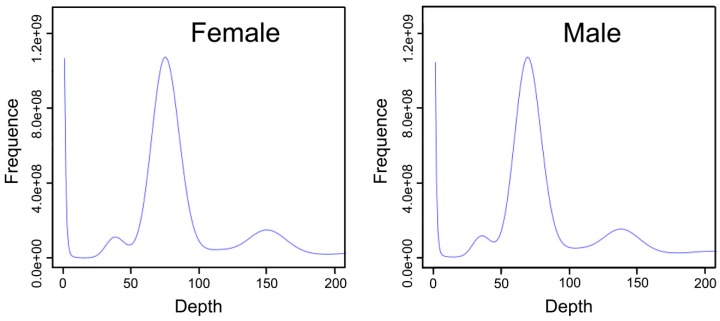
Distribution of 17-mer depth and frequency of female and male *S. argus*. The x-axis indicates depth; the y-axis indicates the proportion representing the frequency at that depth divided by the total frequency of all depths.

**Figure 2 animals-09-01117-f002:**
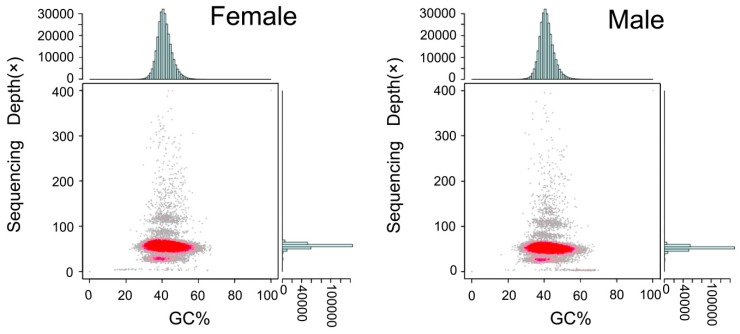
GC content and depth correlation analysis of female and male *S. argus*. The x-axis is the percentage GC content and the y-axis represents sequencing depth. The distribution of sequence depth is on the right side, while the distribution of GC content is at the top.

**Figure 3 animals-09-01117-f003:**
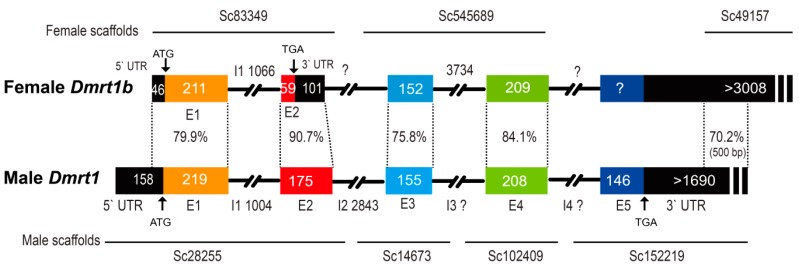
Structure of the *Dmrt1* and *Dmrt1b* genes. *Dmrt1* and *Dmrt1b* are located on the male and female sex chromosomes, respectively. Numbers indicate base pairs (loci) of exon and intron sequences. Percentages indicate the similarity of *Dmrt1* and *Dmrt1b*. Arrows indicate the start and stop codons. Different colored rectangles represent different exons.

**Figure 4 animals-09-01117-f004:**
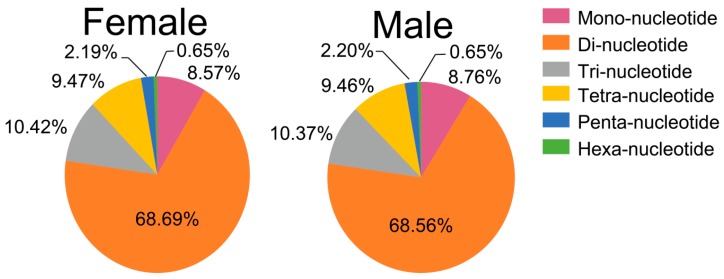
Frequency of SSR types in the genomic survey of female and male *S. argus*.

**Table 1 animals-09-01117-t001:** Statistics of sequencing data of female and male *S. argus*.

Library	Insert Size (bp)	Raw Base (bp)	Effective Rate (%)	Clean Base (bp)	Error Rate (%)	Q20 (%)	Q30 (%)	GC Content (%)
female	350	55,808,601,300	99.80	55,699,379,400	0.03	96.60	91.94	41.50
male	350	51,153,870,900	99.79	51,047,381,700	0.03	96.63	92.03	41.52

**Table 2 animals-09-01117-t002:** Data statistics and analysis of 17-mer.

Identity	K-Mer	K-Mer Depth	K-Mer Number	Genome Size (Mbp)	Revised Genome Size (Mbp)	Heterozygous Ratio (%)	Repeat (%)
female	17	74	45,374,105,016	613.16	598.73	0.37	27.06
male	17	68	41,637,691,628	612.32	597.60	0.38	26.99

**Table 3 animals-09-01117-t003:** Statistics of the assembled *S. argus* genome sequences.

	Identity	Total Length (bp)	Total Number	Max Length (bp)	N50 Length (bp)	N90 Length (bp)
contig	female	580,837,740	444,961	123,323	5,747	590
	male	582,143,644	453,459	110,347	5,745	576
scaffold	female	585,986,615	335,162	231,008	13,556	821
	male	588,188,524	340,134	196,230	13,591	824

**Table 4 animals-09-01117-t004:** Gene function annotation statistics of *S. argus*.

Database	Number (Female/Male)	Percent (Female/Male)
NR	42,825/43,238	45.14%/45.38%
Swissport	33,093/33,359	34.89%/35.01%
KEGG	40,854/41,245	43.07%/43.29%
KOG GO	26,420/26,680 12,428/15,921	27.85%/28.00% 13.10%/16.71%
Annotated	42,869/43,283	45.19%/45.43%
Unannotated	51,993/51,990	54.81%/54.57%
Total	94,862/95,273	100%/100%

**Table 5 animals-09-01117-t005:** SSR (simple sequence repeat) types detected in female and male *S. argus.*

SSR Mining	Total (Female/Male)
Total number of sequences examined	335,162/340,134
Total number of identified SSRs	299,574/299,893
Number of SSR containing sequences	78,202/77,788
Total number of identified SSRs	299,574/299,893
Number of sequences containing more than 1 SSR	39,136/39,104
Number of SSRs present in compound formation	48,384/48,510

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
