# Peer review of "Genome Survey of Male and Female Spotted Scat (Scatophagus argus)"

_animals, 2019, doi:10.3390/ani9121117_

Round 1

Reviewer 1 Report

I was pleased to see that this manuscript has now been revised. The finding about Dmrt1 is interesting, so I would like to expect future synteny analyses.

I have one minor suggestion. Authors said that “Interestingly, the S. argus genomes contained 66.6% and 67.8% of the core genes in conserved actinopterygii orthologs for each sex, respectively.” Why are these findings interesting? I understand that these were caused by incomplete genome assembly based on short NGS reads.

Author Response

Response to Reviewer 1 Comments

Point 1: I was pleased to see that this manuscript has now been revised. The finding about Dmrt1 is interesting, so I would like to expect future synteny analyses.

I have one minor suggestion. Authors said that “Interestingly, the S. argus genomes contained 66.6% and 67.8% of the core genes in conserved actinopterygii orthologs for each sex, respectively.” Why are these findings interesting? I understand that these were caused by incomplete genome assembly based on short NGS reads.

Response 1: Thank you for your positive comments about the manuscript and your suggestions. You are correct that these were caused by incomplete genome assembly based on short NGS reads. "Interestingly" has been removed in the Discussion section in the revised manuscript. The sentence in the Results section now reads "The BUSCO core genes of the male genome were similar to those of the female (Table S3). The partially identified core genes in both sexes were a little high caused by incomplete genome assembly based on short NGS reads.

Reviewer 2 Report

Thank you for your sincere revise. I think this manuscript is sufficient to publish in animals.

Unfortunately, I could not find the supplement data. Would you please attach them?

Author Response

Response to Reviewer 2 Comments

Point 1: Thank you for your sincere revise. I think this manuscript is sufficient to publish in animals.

Unfortunately, I could not find the supplement data. Would you please attach them?

Response 1: Thank you for your positive comments on the manuscript. We have attached all supplemental data. The language was improved by a native English speaker.

This manuscript is a resubmission of an earlier submission. The following is a list of the peer review reports and author responses from that submission.

Round 1

Reviewer 1 Report

In this study, a genomic survey of the spotted scat was conducted using NGS. The provided genomic information will be important for the breeding and aquaculture of S. argus.

However, I have two major concerns.

First is that gene prediction/ annotation has not performed. I think that this type of study should include gene prediction/annotation based on assembled genome, which will be useful information for readers. Second is the finding about Dmrt1.

Although this finding is important and interesting, the methods and results are not well documented. I understood that gene fragments of Dmrt1 was existed in “NGS reads from the male” but not in “NGS reads from the female”. This is not reasonable. If author want to discuss the lack of Dmrt1 in female, gene prediction/annotation and synteny analysis are needed. In addition, completeness assessment of genome sequence is necessary (for example, by BUSCO).

Author Response

Response to Reviewer 1 Comments

Point 1: In this study, a genomic survey of the spotted scat was conducted using NGS. The provided genomic information will be important for the breeding and aquaculture of S. argus. However, I have two major concerns. First is that gene prediction/ annotation has not performed. I think that this type of study should include gene prediction/annotation based on assembled genome, which will be useful information for readers.

Response 1: The suggestion was adopted. The gene prediction and annotation were carried out. We have added gene prediction and annotation into the revised manuscript.

Point 2: Second is the finding about Dmrt1. Although this finding is important and interesting, the methods and results are not well documented. I understood that gene fragments of Dmrt1 was existed in “NGS reads from the male” but not in “NGS reads from the female”. This is not reasonable. If author want to discuss the lack of Dmrt1 in female, gene prediction/annotation and synteny analysis are needed. In addition, completeness assessment of genome sequence is necessary (for example, by BUSCO).

Response 2: Thank you for your positive comments on our work. The suggestion was adopted. Gene prediction and annotation proved that Dmrt1 is male-specific, while Dmrt1b exist in both male and female. As the scaffolds containing Dmrt1 are still short, therefore we cannot carry out synteny analysis at present. We will try to do this in our future study. In addition, completeness assessment of genome sequence was carried out by BUSCO.

Reviewer 2 Report

This manuscript reports the genomic features of the male and female spotted scat based on the draft genome sequences constructed by next generation sequencing technology. As the data is new and the finding will be useful for the development of the spotted scat aquaculture, this report is basically worth publishing in Animals.

However, I think authors should try more advanced analyses, such as the gene annotation and the gene ontology analysis.

The mapping of the genes found in the transcriptome analysis of the spotted scat to the draft genome should be useful to estimate the feasibility of the genome data.

Besides, I think the alignment of the male and female scaffolds will be valuable to explore the mechanisms of the sex differentiation.

Author Response

Response to Reviewer 2 Comments

Point 1: This manuscript reports the genomic features of the male and female spotted scat based on the draft genome sequences constructed by next generation sequencing technology. As the data is new and the finding will be useful for the development of the spotted scat aquaculture, this report is basically worth publishing in Animals. However, I think authors should try more advanced analyses, such as the gene annotation and the gene ontology analysis.

Response 1: Thank you for your positive comments on our draft.

Point 2: However, I think authors should try more advanced analyses, such as the gene annotation and the gene ontology analysis.

Response 2: The suggestion was adopted. We carried out gene annotation and the gene ontology analysis. We used the software GlimmerHMM to predict genes and the predicted genes were used to BLAST with the NR and SwissProt databases using BLASTx (E-value < 1e−5). Meanwhile, Gene ontology (GO), Clusters of euKaryotic Orthologous Groups (KOG), Kyoto Encyclopedia of Genes and Genomes (KEGG) pathway annotations were also assigned to genes using the software Blast2GO software. In addition, the described genes were classified into the KOG slim and GO categories, and then mapped onto the KEGG.

Point 3: The mapping of the genes found in the transcriptome analysis of the spotted scat to the draft genome should be useful to estimate the feasibility of the genome data.

Response 3: Thank you for your suggestion. The reviewer is correct. It is necessary to estimate the feasibility of the genome data. As another Reviewer suggested, we used BUSCO to evaluate our genome sequence. In addition, we mapped the Dmrt1 and Dmt1b found in the transcriptome analysis of the spotted scat to genome data (Supplementary Data). We also analyzed other genes isolated from transcriptome analysis (data not shown). These results confirmed the authenticity of our data.

Point 4: Besides, I think the alignment of the male and female scaffolds will be valuable to explore the mechanisms of the sex differentiation.

Response 4: The suggestion was adopted. We carried out alignment of scaffolds on which we found Dmrt1 exons. We have made it clear in the revised manuscript.
